Optical microscopic study of surface morphology and filtering efficiency of face masks

Neupane Bhanu Bhakta 1 bbneupane@cdctu.edu.np
Mainali Sangita 2
Sharma Amita 3
Giri Basant 3
1 Central Department of Chemistry, Tribhuvan University , Kathmandu , Nepal
2 Department of Chemistry, Amrit Campus, Tribhuvan University , Kathmandu , Nepal
3 Center for Analytical Sciences, Kathmandu Institute of Applied Sciences , Kathmandu , Nepal
Anderson Todd
Electronic publication date: 2019 Jun 26
Publication date: 2019
Volume: 7
Electronic Location ID: e7142
Received 2019 Mar 21; Accepted 2019 May 17
Copyright: © 2019 Neupane et al.
Copyright year: 2019
Copyright holder: Neupane et al.
License: This is an open access article distributed under the terms of the Creative Commons Attribution License, which permits unrestricted use, distribution, reproduction and adaptation in any medium and for any purpose provided that it is properly attributed. For attribution, the original author(s), title, publication source (PeerJ) and either DOI or URL of the article must be cited.
License URL: https://creativecommons.org/licenses/by/4.0/

Keywords: Face mask, Particulate matter, Filtering efficiency, Pore morphology

Funding: Kathmandu Institute of Applied Sciences KIAS2016-1 This work was supported by startup funding to Basant Giri from—Kathmandu Institute of Applied Sciences (No.KIAS2016-1). The funders had no role in study design, data collection and analysis, decision to publish, or preparation of the manuscript.

==============================
Background

Low-cost face masks made from different cloth materials are very common in developing countries. The cloth masks (CM) are usually double layered with stretchable ear loops. It is common practice to use such masks for months after multiple washing and drying cycles. If a CM is used for long time, the ear loops become stretched. The loop needs to be knotted to make the mask loop fit better on the face. It is not clear how washing and drying and stretching practices change the quality of a CM. The particulate matter (PM) filtering efficiency of a mask depends on multiple parameters, such as pore size, shape, clearance, and pore number density. It is important to understand the effect of these parameters on the filtering efficiency.

Methods

We characterized the surface of twenty different types of CMs using optical image analysis method. The filtering efficiency of selected cloth face masks was measured using the particle counting method. We also studied the effects of washing and drying and stretching on the quality of a mask.

Results

The pore size of masks ranged from 80 to 500 μm, which was much bigger than particular matter having diameter of 2.5 μm or less (PM2.5) and 10 μm or less (PM10) size. The PM10 filtering efficiency of four of the selected masks ranged from 63% to 84%. The poor filtering efficiency may have arisen from larger and open pores present in the masks. Interestingly, we found that efficiency dropped by 20% after the 4th washing and drying cycle. We observed a change in pore size and shape and a decrease in microfibers within the pores after washing. Stretching of CM surface also altered the pore size and potentially decreased the filtering efficiency. As compared to CMs, the less frequently used surgical/paper masks had complicated networks of fibers and much smaller pores in multiple layers in comparison to CMs, and therefore had better filtering efficiency. This study showed that the filtering efficiency of cloth face masks were relatively lower, and washing and drying practices deteriorated the efficiency. We believe that the findings of this study will be very helpful for increasing public awareness and help governmental agencies to make proper guidelines and policies for use of face mask.

Introduction

Particulate matter (PM) contributes significantly to overall ambient air pollution. Microscopic particles having a diameter of 2.5 μm or less (PM2.5) can become deposited in the conducting zone of the respiratory system including the alveoli, and penetrate to the cardiovascular system. These fine particles are believed to be responsible for various health problems such as lung inflammation and lung cancer, vascular dysfunction, myocardial infarction, and exacerbation of existing conditions like asthma, diabetes mellitus (Reche et al., 2012; Kim, Kabir & Kabir, 2015; World Health Organization, 2016). The World Health Organization (WHO) estimates that ambient air pollution caused around 4.2 million deaths in 2016, with Asia alone contributing around 60% of total global burden (World Health Organization, 2016).

The PM pollution in many cities globally is higher than the WHO recommended safe limit (Wang et al., 2008; Colbeck, Nasir & Ali, 2010; Sharma et al., 2014; World Health Organization, 2016; Maleki et al., 2016). For example, in Kathmandu Valley, one of the most densely populated and the fastest-growing cities in Asia, air quality is very poor, with PM alone contributing to around 50–60% of total pollution burden. The ambient annual average PM2.5 concentration levels in core urban areas of Kathmandu, Nepal has been reported to be around 54 μg/m3, which is significantly higher than the WHO standard of 10 μg/m3 (World Health Organization, 2016; IQ Air, 2018). In recent years, a systematic study on exposure of PM2.5 (finer fractions of PM10) to subjects of different occupations was made in few core and sub-urban areas of Kathmandu Valley. The hourly personal PM2.5 exposure to traffic police who spend 8–12 h on road duty was found in the range of 40–80 μg/m3 (Gurung & Bell, 2012), which is 1.6–3.2 times higher than WHO standard for 24 h average of 25 μg/m3 (World Health Organization, 2016).

The PM pollution can be minimized by putting forwarding both short- and long-term policies and regulations and enforcing them. Efforts have been made, but their impact in minimizing pollution is almost futile in the cities of developing countries.

Studies have shown that face masks reduce the exposure to PM and other contaminants (Singh et al., 2010; Chughtai, Seale & MacIntyre, 2013; MacIntyre et al., 2015; Shakya et al., 2017). The most commonly used face masks are cloth masks (CMs) and surgical masks (SMs). The effectiveness of a mask is measured by its filtering efficiency. A mask is considered effective if its filtering efficiency is greater than 95%. The efficacy of a mask depends on type of mask material (Mueller et al., 2018), particle size and charge of aerosol tested (Shakya et al., 2017), and user preference (Maxted, 2011; Chughtai, Seale & MacIntyre, 2013).

The CMs are particularly popular in developing countries because they are inexpensive (unit retail price range: USD 0.08–1.5), locally available, and reusable after washing. The CMs are usually double layered (two ply masks) with stretchable ear loops (see Figs. 1A and 1B). The face masks have also been widely used in healthcare facilities to minimize contamination (Chughtai, Seale & MacIntyre, 2013), in emergency situations such as volcanic eruptions (Mueller et al., 2018) and in occupational hazard protection (Belkin, 1997; Van Der Sande, Teunis & Sabel, 2008; MacIntyre & Chughtai, 2015; Cherrie et al., 2018).

Figure 1 Types of mask and survey.

(A) Images of some of the most commonly used cloth masks, and less commonly used surgical mask (B). (C) Face mask use pattern in Kathmandu. The % errors in the figure represent standard deviation.

Previous studies have reported the measurement of filtering efficiency of CMs using lab generated aerosol, polystyrene latex, virus, micro-size particles and diesel exhaust (Rengasamy, Eimer & Shaffer, 2010; MacIntyre et al., 2015; Shakya et al., 2017). As the ambient aerosol we are exposed to is a mixture of particles of varying size and shape in ambient environmental condition, use of simulated particles in controlled laboratory setting may not truly represent the filtering efficiency of masks.

The most commonly used face masks in low-income countries are CMs. Such masks are used for many days and are also used after washing and drying multiple times. In addition, the one-for-all type ear loop of mask does not fit everyone. Therefore, the loop has to be knotted or stretched for better fitting on the face. The washing, drying and stretching practices may change the pore size and porosity of the fabric and thus may deteriorate filtering efficiency. There are no published reports on the surface morphology of face masks and the effect of washing and stretching on pore size of the masks.

In this work, we report on the detail study on surface characterization (pore size, shape, clearance, and pore distribution) of commonly available cloth face masks purchased from local market of Kathmandu, Nepal, and compared with surgical face masks using bright field microscope. We then measured the efficacy of masks on filtering ambient outdoor aerosol particles by particle counting method. Finally, we report on the effect of washing and drying cycles and stretching on surface characteristics and filtering efficiency of CMs.

Materials and Methods

Survey study

We began our study by conducting a survey to know the type of masks people use in Kathmandu. We counted 1,500 people walking through the junction road in Kalanki, Kathmandu and noted number of people wearing face mask and type of mask. The visual counting was carried out from 9.00 am to 1.00 pm for 3 consecutive days in the month of May 2016. Kalanki is a major traffic intersection in Kathmandu.

Surface characterization of masks

A total of 20 different types of cloth face masks (CM) and seven different brands of SMs were purchased from local markets in Kathmandu (Figs. 1A and 1B). The CM types were selected on the basis of design and fabric material. A small section of each mask was cut and was imaged using an optical microscope in bright field mode using objective of 0.4 NA (10×, air) with total magnification of 100×. A built in white LED was used as illumination source and image was acquired by a CMOS camera (AmScope, Irvine, CA, USA). For each mask 10 images were taken and the collected images were processed in ImageJ software (NIH, Bethesda, MD, USA). The field of view of measured image was calibrated by using a calibration glass slide having grid size of 10 μm (AmScope, Irvine, CA, USA). The intensity of light used before the lens was around one mW/cm2. Acquisition time of image, unless specified, was set to 500 m.

Measurement of filtering efficiency

Microscope coverslips (Corning, 1.5, 22 × 22 mm) were placed on a polystyrene petri dish (Microteknic, Haryana, India, 80 mm diameter and 13 mm depth) and the petri dish was covered with a face mask. To make sure coverslip surface is free from particles, the surface was cleaned and inspected by the bright field microscope. To ensure no marginal leakage of particles, the mask was fastened to the petri dish with a rubber band. The whole assembly was kept at 20 feet above the ground in an open box in central Kathmandu. Major sources of PM pollution in this area are vehicular exhausts, dust suspensions, and particles from poorly maintained roads. For control measurement, a second assembly was made without mask and kept in the same place. Both assemblies were exposed to PM for 30 min and number of particles deposited on the glass coverslip surface was counted after the surface was imaged at magnification of 100× with a bright field microscope.

One mask type was sampled at a time and triplicate measurements were carried out for each mask. All experiments were made in sunny days during 11.00 am to 1.00 pm time period. Although we did not measure the wind velocity at our site, the wind velocity was low (∼10 km/h) at Tribhuvan International Airport, which is just two km aerial distance away.

The filtering efficiency was calculated as:(1) Filtering efficiency (%)=(a−b)100/a,

where a and b are the number of particles without and with mask, respectively.

To measure the mask efficiency after washing and drying cycles, mask was soaked for 1 h in 2% (w/v) aqueous solution of powder detergent that contained alkyl-benzene sulfonate and sodium triphosphate as main ingredients. The mask was rinsed multiple times with water so as to get rid of the detergent. The mask was then laid on a flat surface to make sure no stretching of the cloth fibers, and the mask was air dried. Filtering efficiency was measured after each washing and drying cycle by using the procedure mentioned in the above paragraph.

To the best of our knowledge, the particle counting method stated above is novel for the determination of filtering efficiency of a mask.

Determination of particle size

To measure the particle size, particles were deposited on coverslip and imaged at 100× magnification using the bright field microscope. The resolution of the microscopic system was 0.6 μm. The field of view of measured image was calibrated by using a calibration glass slide having grid size of 10 μm (AmScope, Irvine, CA, USA), and particles size was estimated.

Results

Survey study

A survey conducted in Kalanki, Kathmandu, Nepal showed that 31% people used face mask while they were at or close to busy roads (see Fig. 1C). Kalanki is an urban area of Kathmandu and a major cross-section for exit and entry to the Kathmandu Valley from rest of the Nepal. It gets large number of vehicular movements since early morning to late night.

Surface characterization

The representative microscopic images of different CMs and SMs are shown in Fig. 2. Out of 20 CMs (CM1-CM20) imaged, for brevity, images of CM1, CM3, CM7, CM9, CM12, CM18 are shown in A, B, C, D, E, and F, respectively. The bright patches in the image are the pores present in the mask. All CMs studied contained two ply (layers) and the surface characteristics of both layers was very similar.

Figure 2 Bright field microscopic images of mask surfaces.

(A) Representative images for CM1, (B) CM3, (C) CM7, (D) CM9, (E) CM12, and (F) CM18. (G) Representative images of inner, (H) middle, and (I) outer layers of a three layered surgical mask. Scale bar shown (A) is 500 μm and applied to all images.

Although the pore shape and size in CMs were not uniform (see Figs. 2A–2F), we tried to extract quantitative information on the size of the pores by measuring the longest dimension of each pore. Such measurements provided an upper estimation of the size of a pore in each CM. The mean pore size ranged from 81 to 461 μm, with smallest pore size observed for CM4 (81 ± 29 μm) and largest pores observed for CM9 (461 ± 108 μm).

The particulate matter filtering efficiency of a mask also depends on number of pores per unit area; referred here to as pore number density. To get this information, we counted number of pores per microscopic field of view (field of view was 4.5 mm2). We found very diverse number of pores ranging from around 12 (CM11) to 47 (CM15).

For comparison, we also examined the surface of seven different brands of paper/SMs masks available in market. The masks examined contained two or three layers (two or three ply). The surface morphology of the inner, middle, and outer layers of a three ply SM is shown in Figs. 2G–2I, respectively.

Filtering efficiency

To find out if there is any correlation between surface structure and filtering efficiency, we measured the filtering efficiency of selected CMs (CM3, CM7, CM9, CM18), and one SM. The filtering efficiency is reported in Fig. 3A. The filtering efficiency was measured by particle counting method using ambient PM. The particle size information is shown in Fig. 3B. Around 98% particles in ambient air was smaller than 10 μm which is consistent with a recent study reported by our group (Rauniyaar, Aryal & Neupane, 2019). Thus, the filtering efficiency of face mask reported in Fig. 3A can be considered as PM10 filtering efficiency. Figure 3A also shows that filtering efficiency of CMs ranged from 63% to 84%, with the poorest efficiency of 63% measured for CM9. The filtering efficiency of SM was found to be 94%.

Figure 3 Measurement of filtering efficiency.

(A) Filtering efficiency of selected cloth face masks (CM) and a surgical mask (SM). The error bars represent standard deviation in each efficiency value reported (n = 3). (B) Size distribution of ambient particulate matter used in this study.

Stretching effect on mask surface

We also explored how the surface of a mask changes on stretching the mask. To explore this, surface of a CM was stretched and microscopically observed the surface while being stretched. Representative images of mask CM7 in stretched (change in length of mask/original length of mask = ΔL/L ∼0.05) and unstretched conditions is shown in Figs. 4A and 4B, respectively. Careful comparison of these images shows that on stretching surface is distorted, with an increase in pore size and a change in shape. We found similar effects for other CMs, although the extent of the distortion was different. We did not find such changes for SM.

Figure 4 Effect of stretching on mask surface.

(A) Bright field microscopy images of CM7 in unstretched and (B) stretched conditions. Scale bar shown in (A) is 500 μm.

Effect of washing and drying

To explore the effect of washing and drying in the filtering efficiency, we selected a cloth mask CM9. The filtering efficiency measured after each washing and drying cycle for up to four cycles is shown in Fig. 5. With increase in washing and drying cycles there is gradual decrease in filtering efficiency (R2 = 0.99).

Figure 5 Effect of washing and drying on filtering efficiency.

Filtering efficiency for CM9 measured after washing and drying cycles. The error bars represent standard deviation in each efficiency value reported (n = 3).

To find the possible cause of change in filtering efficiency with washing and drying, we imaged mask surface after each cycle. The representative images are shown in Fig. 6.

Figure 6 Optical images after washing and drying cycles.

(A) Bright field microscopic images of unwashed CM9, and after (B) first, (C) second, (D) third, and (E) fourth washing and drying cycles. The scale bar shown in (A) is 500 μm and applied to all images.

Discussion

Surface characterization

It is very obvious from the images shown in Figs. 2A–2F that pore size, shape, inter pore distance, and number of pores per field of view were very different for all the CMs. For examples, CM7 has the smallest and nearly circular pores and CM9 has largest pores and pores are nearly hexagonal, CM12 has medium sized triangular pores. If pores of all CMs are closely inspected, the pores are not perfectly clear but contain microfibers passing from one end of pore to next.

It is evident that the inner (Fig. 2G) and outer (Fig. 2I) layers of surgical face mask have a very similar surface structure, with distinctly visible interwoven cellulose fibers and open spaces. Although the features in the middle layers (Fig. 2H) are not distinctly discernible due to the limited resolution of the microscopic set up used in this study, we can tell that this layer has mocro/nanoporous membrane-like structures. The presence of three layers, with the middle layer having very small pores, shows that the medical mask may be efficient in blocking the PM.

We also explored if there is any difference on the interior surface of CMs and SM. This was achieved by scanning the mask surface in axial direction while keeping the microscope objective fixed. Video was recorded during the axial scanning. A representative video collected for CM and SM is shown in Supplemental Information 1 and 2, respectively. A careful comparison of two videos tells that SM has nicely interwoven microfibers with smaller pores indicating better filtering efficiency in compassion to CMs.

Filtering efficiency

Filtering efficiency is particle size dependent (Belkin, 1997) with lower efficiency observed for smaller sized particles. The size of PM is source dependent and depending on source size can be as small as few tenth of nanometer (Mirowsky et al., 2013). The size distribution of ambient particles used in this work is shown in Fig. 3B. Lateral resolution (Dxy) of microscope system used in this study, based on Eq. (2) (Hell, 2007; Stender et al., 2013; Neupane, 2016)(2) Dxy=λ2NA

with objective of 0.4 NA (NA = numerical aperture) and illumination light of wavelength (λ) of 500 nm, is around 0.6 μm. In this regard, actual size of particles measured can have some uncertainty. Nonetheless, classification of particles in three categories viz. <5, 5–10, and >10 μm was possible in our case. As shown in Fig. 3B, a total of 98% particles were smaller than 10 μm, which is in consistent with a recent study reported by our group (Rauniyaar, Aryal & Neupane, 2019). Thus, the filtering efficiency of the face mask reported in Fig. 3A can be considered as PM10 filtering efficiency.

Bright filed microscopy is frequently used in the determination of number density of micro-meter sized objects (Ricardo & Phelan, 2008; Drey, Graber & Bieschke, 2013; Rauniyaar, Aryal & Neupane, 2019). The accuracy in such measurement is high if: (1) light scattering efficiency of an object to be imaged is high so they can be contrasted from the background, and (2) individual dispersion in the field of view can be achieved. In our case, scattering efficiency of particles is higher than background, and we maintained number of particles per field of view low so that number density can be determined.

The filtering efficiency reported in Fig. 3A was negatively correlated with pore size of the masks, (R2 = 0.94) (Fig. 7). The poor efficiency of CM9 (∼63%) is due to the presence of larger and open pores (pores size 461 ± 113 μm) and improved efficiency of CM7 (∼84%) is due to presence of smaller pores (pores size 100 ± 53 μm). It is interesting to note that filtering efficiency of CM9 is still >60% although the pores are much bigger than the reported particle size. This contradiction may be due to low pore density of the masks and mismatch of the pores in two layers during use. The filtering efficiency of SM was 94%. Again, the excellent filtering efficiency can be attributed to the presence of nicely interwoven microfibers with small pores (in middle layer) as reported in Figs. 2G–2I. The difference in morphology of surgical and CMs is more obvious in Supplemental Information 1 and 2 that show axial scanning of a CM and a SM, respectively.

Figure 7 Correlation between pore size and filtering efficiency.

Each data point represents the mean pore size of mask (in μm) plotted as a function of filtering efficiency. The solid line is the linear fit to the data points.

We used a novel particle counting method to determine the filtering efficiency of face masks. Our method is passive sampling method as it measures the filtering efficiency of free falling dust particles without considering active suction to mimic human inhalation and exhalation. Efficiency was determined by the passive transfer of the particles from one side of the mask to the other. Therefore, this method provides the upper estimate of filtering efficiency or best-case-estimate of capture efficiency of mask. Although our method did not mimic the breathing condition, it allowed us to measure the filtering efficiency using ambient PM. If a standard mask (N95 or 99) is available, our method could be a cheaper alternative to screen the relative efficiency of a mask of unknown efficiency and compare with a standard mask such as N95 or N99.

Filtering efficiency of CM has also been reported in other studies by using active sampling method (Rengasamy, Eimer & Shaffer, 2010; Shakya et al., 2017). They have reported that filtering efficiency of a CM depends on the nature of particles (size and charge) used to measure the filtering efficiency, nature, and design of mask. Their reported efficiency is in the range of 10–90% and conclude that CM perform poorer than N95 SM. In contrast, we used passive method and ambient PM. The surface characteristics of masks could also be different. Therefore, efficiency values are difficult to compare. Nonetheless, our conclusion is same; that is, CM perform poorer than SMs.

Stretching effect on mask surface

Figure 4 shows that mask (CM7) surface changed on stretching with significant increase in pore size in stretched mask. This observation is very important. People use CM for months and the ear loops get stretched. The loop has to be knotted to make the mask fit better on the face. If mask having knotted ear-loop is used, it is very likely that the whole mask surface gets stretched. Although we did not measure the filtering efficiency while the mask was stretched, it can be easily inferred that efficiency will decline if a mask with a knotted ear-loop is used due to changes in pore morphology as observed in Fig. 4.

Effect of washing and drying

The filtering efficiency measured after each washing and drying cycle for up to four cycles is shown in Fig. 5. We found a gradual decrease in filtering efficiency with an increase in washing and drying cycle. As compared to an unwashed mask of efficiency ∼63%, after the 4th washing and drying cycle there was ∼20% drop in filtering efficiency.

A close observation of images in Fig. 6 shows that small changes in surface morphology occur after each cycle. The first change is increase in pore size and change in pore shape. The second change is decrease in number of microfibers within the pore so that pore look more open that is, increase in pore clearance. These changes on the mask should be responsible for decline in efficiency after washing and drying cycle, in consistent with the data reported in Fig. 5.

Conclusions

We studied the effect of surface morphology of locally available face masks on their PM filtering efficiency. Filtering efficiency of CM for ambient PM10 was poorer than in SM. The poor efficiency was due to the presence of larger sized pores. Our study also demonstrated that washing and drying cycle deteriorates the filtering efficiency due to change in pore shape and clearance. We also found that stretching of the CM surface alters the pore size and potentially decreases the filtering efficiency. The findings of this study suggest that CM are not effective, and that effectiveness deteriorates if used after washing and drying cycles and if used under stretched condition. We believe that the findings of this study will be helpful for increasing public awareness among populations of developing countries where such masks are very common, and for policy makers to make and implement basic guidelines for face masks for public use.

Supplemental Information

Supplemental Information 1 Axial scanning of a cloth mask.

A short video that shows scanning of a cloth mask (CM9) at various axial positions.

Click here for additional data file.

Supplemental Information 2 Axial scanning for a surgical mask.

A short video that shows scanning of surgical mask (SM) at various axial positions.

Click here for additional data file.

Supplemental Information 3 Raw data: survey, filtering efficiency, particle size, and pore size data.

Click here for additional data file.

Additional Information and Declarations

Competing Interests

Author Contributions

Data Availability

The authors declare that they have no competing interests.

Bhanu Bhakta Neupane conceived and designed the experiments, analyzed the data, contributed reagents/materials/analysis tools, authored or reviewed drafts of the paper, approved the final draft.

Sangita Mainali conceived and designed the experiments, performed the experiments.

Amita Sharma performed the experiments, prepared figures and/or tables.

Basant Giri analyzed the data, contributed reagents/materials/analysis tools, authored or reviewed drafts of the paper, approved the final draft.

The following information was supplied regarding data availability:

The axial scanning of the cloth mask 9 (CM9) surgical mask (SM), and raw measurements are available in the Supplemental Files.

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
