# Peer review of "Optical microscopic study of surface morphology and filtering efficiency of face masks"

_PeerJ, doi:10.7717/peerj.7142_

## Round 0.1 · original submission · Major Revisions

Please address all of the reviewer comments in your revision. I note in particular the need for a justifying not using a system that pulls air through the masks to simulate human breathing.

Reviewer 1 ·

Basic reporting

no comment

Experimental design

no comment

Validity of the findings

No comment

Additional comments

This manuscript represents the microscopic characterization of the study of surface morphology and filtering efficiency of face masks. The authors tried to quantitatively describe the filtering efficiency of the face mask in connection with washing and drying. The topic is interesting and I recommend for publication in PeerJ with minor revisions as listed below:
(1) Authors expressed pore size in the micrometer scale in their results and compared with PM2.5 and PM10 size. I suggest defining PM2.5 and PM10 size in the micrometer scale so that readers can easily understand the comparison.
(2) % is needed in the y-axis of Figure 3a and x-axis of Figure 7

Reviewer 2 ·

Basic reporting

See attached review.

Experimental design

See attached review.

Validity of the findings

See attached review.

Annotated reviews are not available for download in order to protect the identity of reviewers who chose to remain anonymous.

Reviewer 3 ·

Basic reporting

Overall, this is an interesting study associated with an important subject related to human health – how well do cloth masks often worn by citizens of countries with high air pollution protect them from inhaling airborne particulates? The introduction provides a good rationale for conducting the study and is well supported by scientific literature. The methods are somewhat well described. Additional points are made below. The Discussion needs to be carefully reviewed for results that should be moved to the Results section as mentioned below. The supporting figures were designed well graphically and enhance the outcomes of this study.

Experimental design

Numbers are line numbers in the manuscript

124: If this method for determining filter efficiency is novel then that should be stated. If it is derived from another study, then that study should be cited.

As described, this method is severely limited by not applying active air flow through the mask to replicate breathing through it. If only a passive transfer of particles, primarily by gravity settling of particles through the mask, is measured, then the results obtained would represent a best-case estimate of capture efficiency. This should be discussed in the Discussion as a study limitation.

Likewise, more description is needed regarding the placement of the assembly “at twenty feet above the ground”. Were they placed in an open box or a shelf? Was there any attempt to make the measurements during the same time of day? And, most importantly, the same weather (especially wind) conditions? Were they sampled one-mask-at-a-time or were multiple masks sampled at the same time? These details should be described as they are factors that can greatly influence the amount collected.

143: The magnification used to make these counts should be mentioned, along with a statement of the smallest particle size measurable at that magnification. This detail is later provided in the Discussion (lines 273-278) and should be moved to either the Methods here, or as a Result.

159: the statement that 80% used different types of cloth masks is not supported in Fig 1B which only separates by cloth vs surgical masks. Refer to figure 1B at the end of the first sentence (155) rather than here to fix this problem.

250-251 These values for pore size constitute a result and should have been given in the Results section (line 167 section). Here should just be a discussion of those results and their implications on filtering qualities, and with respect to the results of any other studies that characterized surface qualities.

253 The same is true for the results presented here.

269 This discussion should include a comparison between the efficiency values found here and those found by other researchers, especially that of Rengasamy et al. 2010 who show penetration values as high as the efficiency values shown here, which is a consequence of using a system that provides air flow through the masks versus one that does not (this study).

Validity of the findings

291 Results not presented in the Results section should not be brought up here in the Discussion. State them in Results and discuss them here.

329 A conclusion is provided here: “Findings of this study suggest that cloth mask are not effective.” However there is no basis stated for what is “effective” versus what is not “effective”. For example, somewhere in the Introduction or Methods a statement should be made as to the relationship between mask efficiency and an assessment of effectiveness.

Additional comments

A major issue with this paper is the lack of using a system that pulls air through the masks to simulate human breathing through them. The passive process used is apparently novel and therefore not comparable to results found in other studies. It also provides very high estimates of efficiency considering that a particle has to fall (or waft) through the mask pores rather than be pulled through. The concern, then, is that the results presented here provide a much more favorable indication of the ability of these masks to block the passage of particles through them than would occur when worn (and assuming a good fit against the face, which wasn’t discussed). If those results remain, the authors must provide a thorough comparison of these results with those of others with the warning that these results do not represent the efficiency when breathed through. Instead the results are just a general comparison of relative performance between mask types.

As such, it would be preferred that the plot associated with Figure 3A not be shown. Instead, the other figures can be retained (including Fig 5) so that the point of this paper is to demonstrate how cloth masks compare structurally to surgical masks, and how they compare before and after stretching and washing.

---

## Round 0.2 · accepted · Accept

Thank you for your efforts in revising your manuscript to address reviewer concerns.

Reviewer 1 ·

Basic reporting

No comment.

Experimental design

No comment.

Validity of the findings

No comment.

Additional comments

The authors addressed my comments in the revised version of the manuscript. Therefore, I recommend the paper for publication in this journal.

Reviewer 2 ·

Basic reporting

The authors have satisfactorily addressed my concerns and improved the manuscript with clarification and suggested changes.

Experimental design

No comment

Validity of the findings

No comment